# Towards Hardware-Aware Tractable Learning of Probabilistic Models

**Laura I. Galindez Olascoaga**[†]**, Wannes Meert**[‡] **, Nimish Shah**[†]
**Marian Verhelst**[†] **and Guy Van den Broeck**[††]
[†] Electrical Engineering Department, KU Leuven
[‡] Computer Science Department, KU Leuven
[††] Computer Science Department, University of California, Los Angeles
{laura.galindez,nimish.shah,marian.verhelst}@esat.kuleuven.be
wannes.meert@cs.kuleuven.be, guyvdb@cs.ucla.edu

## Abstract

Smart portable applications increasingly rely on edge computing due to privacy and latency concerns. But guaranteeing always-on functionality comes with two major challenges: heavily resource-constrained hardware; and dynamic application conditions. Probabilistic models present an ideal solution to these challenges: they are robust to missing data, allow for joint predictions and have small data needs. In addition, ongoing efforts in the field of tractable learning have resulted in probabilistic models with strict inference efficiency guarantees. However, the current notions of tractability are often limited to model complexity, disregarding the hardware's specifications and constraints. We propose a novel resource-aware cost metric that takes into consideration the hardware's properties in determining whether the inference task can be efficiently deployed. We use this metric to evaluate the performance versus resource trade-off relevant to the application of interest, and we propose a strategy that selects the device settings that can optimally meet users' requirements. We showcase our framework on a mobile activity recognition scenario, and on a variety of benchmark datasets representative of the field of tractable learning and of the applications of interest.

## 1 Introduction

Tractable learning aims to balance the trade-off between how well the resulting models fit the available data and how efficiently queries are answered. Most implementations focus on maximizing model performance and only factor in query efficiency by subjecting the learning stage to a fixed tractability constraint (e.g. max treewidth [2]). While recent notions of tractability consider the cost of probabilistic inference as the number of arithmetic operations involved in a query [27, 28], they still disregard hardware implementation nuances of the target application. This is of special concern for edge computing on embedded applications, where the target algorithm must run in resource constrained hardware, such as a small ARM or RISC-V embedded processor, or a microcontroller. For such architectures running a lightweight operating system, the overall compute cost is mostly determined by the cost of fundamental arithmetic operations, the interaction with sensor interfaces and the device's memory transactions [18, 12].

In addition, efforts towards hardware-efficient realizations of probabilistic inference are currently scarce [37, 22, 35]. This is in stark contrast with the tremendous progress achieved by embedded neural network implementations [38, 19, 30] .

We address these limitations of the field of tractable learning by proposing a novel resource-aware cost metric that takes into consideration the target embedded device's properties (e.g. energy consumption);

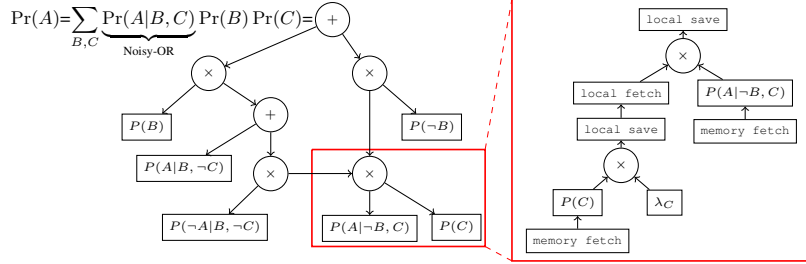

Figure 1: Arithmetic Circuit from a compiled noisy-OR and its mapping to hardware.

and system-level configuration (e.g. sensors used). We map these hardware characteristics to the cost vs. performance trade-off space, and propose a set of techniques to find the optimal system-level configuration. Specifically, we address the following points: (a) Section 3 discusses the relevant hardware-aware tractability metrics, and Section 4 defines the problem statement; (b) Section 5 discusses how to exploit the model's properties to exchange task-performance for hardware efficiency, and introduces techniques to find the optimal set of system configurations in the cost vs. performance trade-off space; and (c) Section 6 shows practical examples of these optimal solutions. This work constitutes one of the first efforts to introduce the field of tractable probabilistic reasoning to the emerging domain of edge computing. This is motivated by probabilistic models' traits, several of which are ideal for portable applications that require reasoning on the edge: robustness to missing information, small data needs, joint predictions, and expert knowledge integration. Moreover, unlike fixed neural architecture training, tractable learning allows to explicitly vary the level of complexity of the inference task, which allows us to model resource tunability.

## 2   Background and motivation

We use standard notation: random variables are denoted by upper case letters $X$ and their instantiations by lower case letters $x$. Sets of variables are denoted in bold upper case $\mathbf{X}$ and their joint instantiations in bold lower case $\mathbf{x}$. Sets of variable sets are denoted with $\mathcal{X}$.

The model representation of choice in this paper is the Arithmetic Circuit (AC), a state-of-the-art, compact representation for a variety of machine learning models such as probabilistic graphical models (PGMs) [6] and probabilistic programs [10]. Recent developments show how the structure of ACs can also be learned from data [25, 24]. Furthermore, ACs can be complemented with deep learning architectures [42, 29] to achieve the best of both worlds. An alternative representation of ACs are Sum-Product Networks (SPNs), which can also encode probability distributions as a computational graph [33, 14]. SPNs can be efficiently converted to ACs and vice versa [34].

### 2.1   Probabilistic inference with Arithmetic Circuits

An AC is a directed acyclic graph where inner nodes represent addition or multiplication and leaf nodes are real-valued variables. ACs constitute a standard representation for computing polynomials, but they have proven to be efficient for reasoning over knowledge bases and probabilistic models when a number of additional properties are enforced on them [6]. Once the circuit is known, the complexity of executing the encoded formula is also known, since marginalization and partition function operations are polynomial in the size of the circuit [4], thus making them a well-suited representation for tractable learning. ACs represent a joint probability distribution over a set of random variables $\mathbf{X}$: the leaf nodes are either binary indicator variables $\lambda_{X=x}$, where $X \in \mathbf{X}$, or parameters $\theta$. Figure 1 shows an example of an AC that encodes the joint probability distribution of a noisy-OR model [16].

This representation allows to perform inference to answer a number of probabilistic queries. For example, given an instantiation $\mathbf{f}$ of $\mathbf{F} \subseteq \mathbf{X}$, the marginal probability $\Pr(\mathbf{f})$ can be computed by setting the indicator variables to 1 if they correspond to instantiations consistent with the observed values, $\lambda_{X=x} \leftarrow 1_{x \sim \mathbf{f}}$, and subsequently performing an upward pass on the AC [4]. In a binary classification task, one can define a class variable $C$, a feature set $\mathbf{F}$ and a classification threshold $T$, assumed to be $0.5$ in this work. For a given instance $\mathbf{f}$, the task consists of selecting the class $C_T$

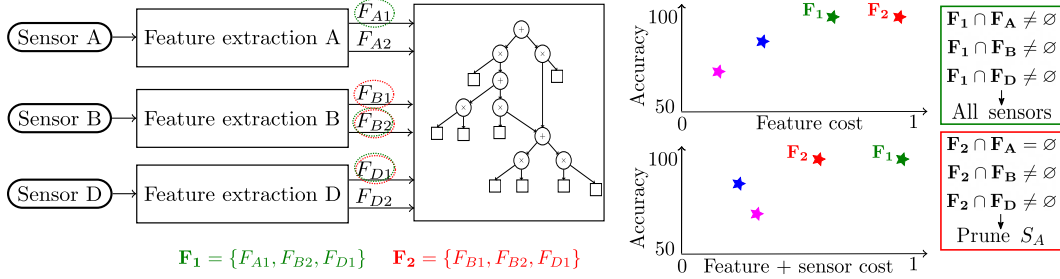

Figure 2: Sensory embedded classification example.

for which the condition $Pr(C|\mathbf{f}) \geq 0.5$ is met. The conditional probability can be calculated by performing two upward passes on the AC[1] that encodes $\Pr(C, \mathbf{F})$, after setting the indicator variables $\lambda$ in accordance to instance $\mathbf{f}$. ACs' straightforward mapping to embedded hardware and the fact that they readily encode the algorithm necessary for inference, motivates our choice for this probabilistic model representation. Moreover, the process of learning them already entails a trade-off between their predictive performance and their computational efficiency. The following section motivates our proposed hardware-aware tractability metric.

## 2.2 Motivating example

Consider the mobile classification scenario in Figure 2, where the feature set $\mathbf{F}=\{F_{A1},F_{A2},F_{B1},F_{B2},F_{D1},F_{D2}\}$ is extracted from sensors $A$, $B$ and $D$, and where the AC is assumed to be the most compact model that maximizes classification accuracy. Suppose there are two feature subsets available, $\mathbf{F_1}=\{F_{A1}, F_{B2}, F_{D1}\}$ and $\mathbf{F_2}=\{F_{B1}, F_{B2}, F_{D1}\}$; and that they attain the same accuracy. Hence, the goal is to find the least expensive subset. The solution to this problem would clearly be $\mathbf{F_1}$ when considering only feature cost, a common approach to address the problem of feature subset selection. But when considering also the costs of the sensors, $\mathbf{F_2}$ turns out to be a better choice, as sensor $A$ is unused and can be turned off. This example shows that trade-off opportunities can be missed when failing to describe realistic hardware-aware system-level costs.

## 3 Hardware-aware cost

In this section we formalize the notion of hardware-aware cost, the basis of our optimization framework. Let $\alpha = \langle +, \times, \theta \rangle$ be an AC that encodes a joint probability distribution over variables $\mathbf{F}$, extracted from the set of sensor interfaces $\mathbf{S}$. The *hardware-aware cost* ($\mathrm{C_{HA}}$) is defined as:

$$\mathrm{C_{HA}}(\alpha, \mathbf{S}, \mathbf{F}) = \mathrm{C_{AC}}(\alpha) + \sum_{S \in \mathbf{S}} \mathrm{C_{SI}}(S, \mathbf{F}_S), \qquad (1)$$

where $\mathrm{C_{AC}}$ are the computation costs, pertaining to inference on arithmetic circuit $\alpha$, $\mathrm{C_{SI}}$ are the sensor interfacing and feature extraction costs, and $\mathbf{F}_S$ is the feature subset extracted from sensor $S$.

**Computation costs.** At a high level, a typical embedded hardware architecture entails two components: an on-chip main memory (typically SRAM), which commonly houses the algorithm's parameter set; and a processing unit, where operations are performed and intermediate values are cached in a local memory. Performing an upward pass on an AC involves the following actions (see Figure 1): 1) fetching parameters from the main memory, 2) performing arithmetic operations, consisting of additions and multiplications, 3) caching intermediate values in a local memory (e.g. register file or low level cache) and 4) fetching intermediate values from local memory, as needed.[2] Each action has a significantly different hardware resource cost. For example, post synthesis energy models of a simple embedded CPU show that multiplications can require 4 times as much energy

as additions, and memory transactions 5 times as much energy as multiplications [18]. When it comes to the design of embedded hardware, energy efficiency is indeed one of the main challenges to address. Hence, we continue to focus on this resource as a proof of concept without loss of generality; examples of other relevant hardware resource metrics are throughput and latency. It is evident that the total hardware cost of performing a pass on an AC must factor in all the aforementioned transactions. Let $nb$ be the number of bits used to represents parameters $\theta$ and perform arithmetic operations $+$ and $\times$. The *computation cost* ($\mathrm{C_{AC}}$) of AC $\alpha$ is defined as:

$$\mathrm{C_{AC}}(\alpha, nb) = \mathrm{C_+}(nb) + \mathrm{C_\times}(nb) + \mathrm{C_{mem}}(nb) + \mathrm{C_{cache}}(nb), \tag{2}$$

where the terms in $\mathrm{C_{AC}}$ define the cost incurred by each type of operation. Here, $\mathrm{C_+}$ and $\mathrm{C_\times}$ are the costs of addition and multiplication; $\mathrm{C_{mem}}$ is the cost from fetching parameter leaf nodes from main memory and $\mathrm{C_{cache}}$ is the cost from storing and fetching from local cache (as in Figure 1):

$$\mathrm{C_+}(nb) = \sum_a [a =_t +] \cdot \phi_+(nb), \qquad \mathrm{C_{mem}}(nb) = \sum_a [a \neq_t + \text{ and } a \neq_t \times] \cdot \phi_{mem}(nb),$$
$$\mathrm{C_\times}(nb) = \sum_a [a =_t \times] \cdot \phi_\times(nb), \qquad \mathrm{C_{cache}}(nb) = \sum_a [a =_t + \text{ or } a =_t \times] \cdot \phi_{cache}(nb),$$

where $a$ denotes a node in $\alpha$, the equality $=_t$ holds when node $a$ matches the operation type on the right side and $[\beta]$ is equal to 1 when $\beta$ is true. The function $\phi(.)$ describes the effective cost of the particular operation and can be derived from empirical benchmarks, customized to the target hardware [18, 36]. When expressing cost in terms of energy consumption, computation costs scale with the precision in number of bits used to represent parameters and perform arithmetic operations ($nb$), which is typically the same for all nodes in the AC. To conclude, the cost incurred by each node in an AC is determined by its type (whether addition, multiplication, local parameter fetch, or remote memory access) and the resolution of the operation or parameter (in $nb$).

**Sensor interfacing costs.** The computational block described above is often part of a larger system, which repeatedly performs a task based on external inputs or observations, such as classification. In this scenario, one must factor in the costs incurred by interfacing with the environment or the user. A sensory interface consists of a set of sensors $\mathbf{S}$, which gather, process and digitize environmental information (typically in the analog domain), and a (typically digital) feature extraction unit, which generates the feature set $\mathbf{F}$ to be used by the machine learning algorithm. Let $\mathbf{S}$ be the set of available sensors and $\mathbf{F}$ the feature set extracted from them. The sensor interfacing cost ($\mathrm{C_{SI}}$) is:

$$\mathrm{C_{SI}}(S, \mathbf{F}_S) = \mathrm{C_S}(S) + \sum_{F_S \in \mathbf{F}_S} \mathrm{C_F}(F_S), \tag{3}$$

where $\mathrm{C_S}$ describes the cost incurred by sensor $S$ and $\mathrm{C_F}$ the cost of extracting feature set $\mathbf{F}_S \subseteq \mathbf{F}$. The sensing cost function $\mathrm{C_S}$ can be customized to the target platform and applications through measurements or data sheets. Note that, if no features from a given sensor are used, it can be shut down, and its cost dropped (see Figure 1). In most systems, $\mathrm{C_F}$ can be defined from the type and number of arithmetic and memory operations involved, in a similar fashion to the computation cost estimation $\mathrm{C_{AC}}$, as will be illustrated in the experiments (Section 6.1).

## 4 Problem statement

We have seen so far that $\mathrm{C_{HA}}$ depends on four system properties:1) the complexity of model $\alpha$, determined by the number and type of its operations; 2) the size and type of the feature set $\mathbf{F}$; 3) the size and type of the available sensor set $\mathbf{S}$; and 4) the number of bits $nb$ used within $\alpha$. We refer to an instantiation of these four properties $\sigma = \{\alpha, \mathbf{F}, \mathbf{S}, nb\}$ as a *system configuration*. Clearly, the system configuration also determines the algorithm's performance, defined according to the application of interest. The methods proposed in this work can accommodate any performance metric or miss-classification cost, but we will only consider accuracy, due to its generality. Specifically, we set the classification threshold to $T = 0.5$, and we consider the accuracy of the Bayes-optimal predictions ($Acc$) over a set of feature instantiations $\{\mathbf{f}_1, ..., \mathbf{f}_l\}$.

Section 2.2 asks to identify the system configuration that incurs the lowest cost for a desired accuracy. Similarly, we might be interested in the configuration that achieves the highest accuracy for a given cost constraint. Thus, the problem we aim to address is how to select the system configurations that map to the Pareto-frontier on the hardware-cost vs. accuracy trade-off space. The inputs to our problem are the class variable $C$, the available features $\mathbf{F}$ and sensors $\mathbf{S}$ sets, and the set of available precisions $\mathbf{nb}$. The output is the set of Pareto-optimal system configurations $\sigma^* = \{\{\alpha_i^*, \mathbf{F}^*_i, \mathbf{S}^*_i, nb_i^*\}_{i=1:p}\}$.

# 5 Trade-off space search

We propose to search the cost vs. accuracy trade-off space by scaling four properties (see Section 4):

**Model complexity scaling.** We learn a set of ACs $\boldsymbol{\alpha}$ of increasing complexity. Each maps to a specific classification accuracy and computation cost $C_{AC}$ (see Eq. 2). Although discriminative AC learners have shown state-of-the-art classification accuracy [25], we have opted for a generative learning strategy: the LearnPSDD algorithm [24]. The motivation for this choice is twofold: this algorithm improves the model incrementally, but each iteration already leads to a *valid* AC, that can be used to populate the set $\boldsymbol{\alpha}$. Moreover, the learned ACs encode a joint probability distribution, which ensures they are robust to missing data, as demanded by the application range of interest: embedded reasoning tasks must often deal with missing values, either due to malfunction (e.g., a sensor is blocked in an autonomous driving system), or to enforce hardware-cost efficiency (e.g., when energy consumption is excessive, the driving system has the choice to turn off an expensive sensor and the features extracted from it).

**Feature and sensor set scaling.** We scale the feature set $\mathbf{F}$ by sequentially pruning individual features (see Section 5.1). The effect of feature pruning on classification accuracy has been discussed in numerous works [11, 5, 23] and the impact on the hardware-aware cost is clear from Eq. 3. Pruning features can also have an impact on the computation costs $C_{AC}$: if a variable is always unobserved, its indicator variables are fixed (see Section 2.1), which we exploit to simplify the circuit (see Algorithm 2). In addition, sensor $S \in \mathbf{S}$ can be pruned when none of the features it originates is used anymore; a strategy that has not been explored by the state of the art, but that is straightforward with our approach, since it considers the full system.

**Precision scaling.** We consider four different standard IEEE 754 floating point representations, as they can be implemented in almost any embedded hardware platform. Reducing the precision of arithmetic operations and numerical representations entails information loss and results in performance degradation [36]. The effect on computation costs $C_{AC}$ is clear from Eq. 2.

## 5.1 Search strategy

Finding the smallest possible AC that computes a given function is $\Sigma_2^p$-hard [3], thus computationally harder than NP. No single optimal solution is known for this problem; it is a central question in the field of knowledge compilation [7]. Optimizing for the lowest-cost/highest-accuracy AC, further increases complexity. We therefore opt for a greedy optimization algorithm. Specifically, we rely on a series of heuristics to search the trade-off space. In each step, we independently scale one of the available configuration properties $\langle \alpha, \mathbf{F}, \mathbf{S}, nb \rangle$, as described in the previous section, and aim to find its locally optimal setting. The search begins by learning the model set $\boldsymbol{\alpha} = \{\alpha_k\}_{k=1:n}$. Then, as shown by Algorithm 1, starting from each model $\alpha_k$, we perform a greedy neighborhood search that aims to maximize cost savings and minimize accuracy losses by sequentially pruning the sets $\mathbf{F}$ and $\mathbf{S}$, and simplifying $\alpha_k$ accordingly (Algorithm 2). At each iteration, we evaluate the accuracy and cost of $m$ feature subset candidates, where each considers the impact of removing a feature from the user-defined prunable set $\mathbf{F}_{prun} \subseteq \mathbf{F}$. We then select the feature and sensor subsets $\mathbf{F}_{sel} \subseteq \mathbf{F}$, $\mathbf{S}_{sel} \subseteq \mathbf{S}$ and the simplified model $\alpha_{sel}$ that maximizes the objective function $\text{OF} = acc/cost_{norm}$, where $cost_{norm}$ is the evaluated hardware-aware cost $C_{HA}$, normalized according to the maximum achievable cost (from the most complex model available $\alpha_n$). Note that feature subset selection drives sensor subset selection $\mathbf{S}_{sel}$, as described before, and defined in lines 12 and 19 of Algorithm 1.

The output of Algorithm 1, $\langle \mathcal{F}^{(k)}, \mathcal{S}^{(k)}, \mathcal{M}^{(k)} \rangle$, is a set of system configurations of the form $\{\{\mathbf{F}_{sel,1}, \mathbf{S}_{sel,1}, \alpha_{sel,1}\}, \ldots, \{\mathbf{F}_{sel,q}, \mathbf{S}_{sel,q}, \alpha_{sel,q}\}\}$, where $q = |\mathbf{F}_{prunable}|$, and the superscript $(k)$ denotes the number of the input model $\alpha_k$, taken from $\boldsymbol{\alpha}$. For each configuration resulting from Algorithm 1, we can sweep the available precision configurations $\mathbf{nb}$, for a final space described by $\boldsymbol{\sigma} = \langle \mathcal{F}, \mathcal{S}, \mathcal{M}, \mathcal{N} \rangle$ of size $|\boldsymbol{\alpha}| \cdot |\mathbf{F}_{prunable}| \cdot |\mathcal{N}|$, where $\mathcal{N}$ contains the selected precision. In the experimental section we show a work-around to reduce search space size and the number of steps required by the Pareto-optimal search. Regarding complexity, the feature selection in Algorithm 1 is a greedy search, thus its complexity is linear in the number of features times the number of iterations needed for convergence. [3] The AC pruning routine consists of an upward pass on the AC and its complexity is therefore linear in the size of the AC.

**Algorithm 1:** ScaleSI($\alpha_k$, $\mathbf{F}_{prun}$, $\mathbf{S}_{prun}$)

**Input:** $\alpha_k$: the $k$th model in $\boldsymbol{\alpha}$ , $\mathbf{F}_{prun}$, $\mathbf{S}_{prun}$: set of prunable features and sensors.
**Output:** $\langle \mathcal{F}^k, \mathcal{S}^k, \mathcal{M}^k \rangle$, **acc**, **cost**: $k$th collection of pruned features, sensors and model sets, their accuracy and cost.

1 $\mathbf{F}_{sel} \leftarrow \mathbf{F}_{prun}$, $\mathbf{S}_{sel} \leftarrow \mathbf{S}_{prun}$, $\alpha_{sel} \leftarrow \alpha_k$
2 $\langle \mathcal{F}^k, \mathcal{S}^k, \mathcal{M}^k \rangle \leftarrow \langle \mathbf{F}_{sel}, \mathbf{S}_{sel}, \alpha_{sel} \rangle$
3 $acc_{sel}$=Acc($\alpha_{sel}$, $\mathbf{F}_{sel}$), $cost_{sel}$=$\mathrm{C}_{\mathrm{HA}}(\alpha_{sel}, \mathbf{F}_{sel}, \mathbf{S}_{sel})$
4 $\langle \mathbf{acc}, \mathbf{cost} \rangle \leftarrow \langle acc_{sel}, cost_{sel} \rangle$
5 **while** $|\mathbf{F}_{sel}| > 1$ **do**
6     $ob_{max} \leftarrow 0$     // Initialize objective value
7     **foreach** $F \in \mathbf{F}_{prun}$ **do**
8        $\mathbf{F}_{ca} \leftarrow \mathbf{F}_{sel} \setminus F$
9        $\mathbf{S}_{ca} \leftarrow \mathbf{S}_{sel}$
10        **foreach** $S \in \mathbf{S}_{prun}$ **do**
11           **if** $\mathbf{F}_{ca} \cap \mathbf{F}_S = \varnothing$ **then**
12              $\mathbf{S}_{ca} \leftarrow \mathbf{S}_{ca} \setminus S$     // Prune sensor
13        $\alpha_{ca} \leftarrow$ PruneAC($\alpha_{sel}$, $\mathbf{F}_{ca}$)
14        $acc_{ca} \leftarrow$ Acc($\alpha_{ca}$, $\mathbf{F}_{ca}$)
15        $cost_{ca} \leftarrow \mathrm{C}_{\mathrm{HA}}(\alpha_{ca}, \mathbf{F}_{ca}, \mathbf{S}_{ca})$
16        $ob_{ca} \leftarrow$ OF($acc_{ca}$, $cost_{ca}$)
17        **if** $ob_{ca} > ob_{max}$ **then**
18           $ob_{max} \leftarrow ob_{ca}$
19           $\mathbf{F}_{sel} \leftarrow \mathbf{F}_{ca}$, $\mathbf{S}_{sel} \leftarrow \mathbf{S}_{ca}$, $\alpha_{sel} \leftarrow \alpha_{ca}$
20           $acc_{sel} \leftarrow acc_{ca}$, $cost_{sel} \leftarrow cost_{ca}$
21     $\mathcal{F}^k$.insert($\mathbf{F}_{sel}$), $\mathcal{S}^k$.insert($\mathbf{S}_{sel}$), $\mathcal{M}^k$.insert($\alpha_{sel}$)
22     **acc**.insert($acc_{sel}$), **cost**.insert($cost_{sel}$)
23 **return** $\langle \mathcal{F}^k, \mathcal{S}^k, \mathcal{M}^k \rangle$, **acc**, **cost**

---

**Algorithm 2:** PruneAC($\alpha$, $\mathbf{F}$)

**Input:** $\alpha$: the input AC, $\mathbf{F}$: the observed feature set used to guide the pruning of $\alpha$.
**Output:** $\alpha_{pr}$: the pruned AC.

1 $\alpha_{pr} \leftarrow$ copy($\alpha$)
    /* Loop through AC, children before parents   */
2 **foreach** $a$ *in* $\alpha_{pr}$ **do**
3     **if** *a is an indicator variable* $\lambda_{F=f}$ *and* $F \notin \mathbf{F}$ **then**
4        replace $a$ in $\alpha_{pr}$ by a constant 1
5     **else if** *a is* $+$ *or* $\times$ *with constant children* **then**
6        replace $a$ in $\alpha_{pr}$ by an equivalent constant
7 **return** $\alpha_{pr}$

---

**Algorithm 3:** GetPareto($\boldsymbol{\sigma}$, acc, cost)

**Input:** $\boldsymbol{\sigma}$, **acc**, **cost**: Configuration set, their accuracy and cost.
**Output:** $\boldsymbol{\sigma}^*$, **acc**$^*$, **cost**$^*$: Pareto optimal configurations, their accuracy and cost.

1 $\langle \mathbf{cost}^*, \boldsymbol{\sigma}^*, \mathbf{acc}^* \rangle \leftarrow \langle \{\}, \{\}, \{\} \rangle$ ;
    /* Sort according to ascending cost      */
2 $\langle \mathbf{cost}, \boldsymbol{\sigma}, \mathbf{acc} \rangle \leftarrow sorted(\langle \mathbf{cost}, \boldsymbol{\sigma}, \mathbf{acc} \rangle)$;
3 $i \leftarrow |\boldsymbol{\sigma}| + 1$;
4 **while** $i > 0$ **do**
5     $i \leftarrow \arg\max \mathbf{acc}_{0:i}$
6     $\boldsymbol{\sigma}^*$.insert($\sigma_i$)
7     $\mathbf{acc}^*$.insert($\mathbf{acc}_i$)
8     $\mathbf{cost}^*$.insert($\mathbf{cost}_i$)
9     $i \leftarrow i - 1$
10 **return** $\boldsymbol{\sigma}^*$, **acc**$^*$, **cost**$^*$

## 5.2 Pareto-optimal configuration selection

Algorithm 3 describes how we extract the Pareto-optimal configuration subset, but any convex hull algorithm can be used. The input is the configuration set $\boldsymbol{\sigma}=\langle \mathcal{F}, \mathcal{S}, \mathcal{M}, \mathcal{N} \rangle$ and their corresponding accuracy (**acc**) and cost (**cost**). The output of this algorithm is the set of Pareto-optimal system configurations $\boldsymbol{\sigma}^*=\{\{\alpha_i^*, \mathbf{F}^*_i, \mathbf{S}^*_i, nb_i^*\}_{i=1:p}\}$, each guaranteed to achieve the largest reachable accuracy for any given cost; or the lowest reachable cost for any given accuracy (**acc**$^*$, **cost**$^*$).

Note that the framework introduced thus far can balance the trade-off between hardware-aware cost and any other application-specific performance metric, by simply replacing the accuracy terms in Algorithms 1, 2 and 3 with such a metric. For instance, medical applications often aim to balance precision and recall, and may favor the latter at night under scarce medical supervision. Furthermore, our framework can be used for density estimation tasks by deploying the model complexity scaling followed by precision scaling stages and forgoing the pruning stages of Algorithms 1 and 2, in order to keep the full joint distribution [13]. The next section illustrates how our methodology can reap the benefits of scalable embedded hardware.

## 6 Experimental evaluation

We empirically evaluate the proposed techniques on a relevant embedded sensing use case: the Human Activity Recognition (HAR) benchmark [1]. Additionally, we show our method's general applicability on a number of other publicly available datasets [8, 15, 21, 26, 31], two of them commonly used for density estimation benchmarks and the rest commonly used for classification (see Table 1).[4]

**Computational costs.** The computation costs $\mathrm{C}_{\mathrm{AC}}$ are based on the energy benchmarks discussed in [18] and [36]. Table 2 shows the relative costs of each term in $\mathrm{C}_{\mathrm{AC}}$ and how they scale with

precision $nb$. The baseline is 64 floating point bits because it is the standard IEEE representation in software environments. For the rest of the experiments, we consider three other standard low precision representations: 32 bits (8 exponent and 24 mantissa), 16 bits (5 exponent and 11 mantissa) and 8 bits (4 exponent and 4 mantissa) [20].

**Dataset pre-processing.** For the classification benchmarks, we discretized numerical features using the method in [9]. We then binarized them using a one-hot encoding and subjected them to a 75%-train, 10%-validation and 15%-test random split. For the HAR benchmark, we preserved the timeseries information by using the first 85% samples for training and validation and the last for testing. For the density estimation datasets, we used the splits provided in [26] and we assumed the last feature in the set to be the class variable. On all datasets, we performed wrapper feature selection (evaluating the features' value on a Tree Augmented Naive Bayes classifier) before going through the hardware-aware optimization process to avoid over-fitting on the baseline model and ensure it is a fair

Table 1: Experimental datasets
†: Classification , ⋆: Density est.

| Dataset | $|\mathbf{F}|$ | $|\mathbf{F}_{prun}|$ | $|\boldsymbol{\alpha}|$ |
|---|---|---|---|
| Banknote[†] | 15 | 15 | 11 |
| HAR [†] | 28 | 28 | 11 |
| Houses [†] | 36 | 20 | 11 |
| Jester [⋆] | 99 | 20 | 11 |
| Madelone [†] | 20 | 20 | 11 |
| Nltcs [⋆] | 15 | 15 | 11 |
| Six-HAR [†] | 54 | 20 | 11 |
| Wilt [†] | 11 | 11 | 11 |

reference point. The number of effectively used features $|\mathbf{F}|$ is shown in Table 1. In addition, we consider all the features to be in the prunable set $\mathbf{F}_{prun}$ for datasets with less than 30 features. For the rest, we consider the 20 with the highest correlation to the class variable. Within the context of an application, the prunable set can be user-defined. For instance, in a multi-sensor seizure detection application, medical experts might advise against pruning features extracted from an EEG monitor.

**Model learning.** We learned the models on the train and validation sets with the LearnPSDD algorithm [24], using the same settings reported therein, and following the bottom-up vtree induction strategy. To populate the model set $\boldsymbol{\alpha}$, we retained a model after every $N/10$ iterations, where $N$ is the number of iterations needed for convergence (this is until the

Table 2: Experiment computational costs.

| Operation | At 64 bits | Operation cost |
|---|---|---|
| $C_{mem}$ | 1 | $\phi_{mem} = \gamma_{mem} \cdot nb$ |
| $C_{cache}$ | 0.2 | $\phi_{cache} = \gamma_{cache} \cdot nb$ |
| $C_{\times}$ | 0.6 | $\phi_{\times} = \gamma_{\times}^2 \cdot nb^2 \cdot \log(nb)$ |
| $C_{+}$ | 0.1 | $\phi_{+} = \gamma_{+} \cdot nb$ |

log-likelihood on validation data stagnates). Table 1 shows $|\boldsymbol{\alpha}|$ for each dataset. Furthermore, as a baseline, we trained a Tree Augmented Naive Bayes (TAN) classifier and compiled it to an AC.[5]

## 6.1 Embedded Human Activity Recognition

The HAR dataset aims to recognize the activity a person is performing based on statistical and energy features extracted from smartphone accelerometer and gyroscope data. We perform binary classification by discerning "walking downstairs" from the other activities. For the experiments, we use a total of 28 binary features, 8 of which are extracted from the gyroscope's signal and the rest from the accelerometer, as detailed in Appendix B.4. All computation costs for this dataset are normalized according to the energy consumption trends of an embedded ARM M9 CPU, assuming 0.1nJ per operation [39]. Sensors are estimated to consume 2mWatt, while the costs of all features is defined as 30 MAC operations (see Appendix B.1 for more details).

**Pareto optimal configuration.** This experiment consisted of three stages, performed on the training set (Figure 3(a)): 1) We first mapped each model in $\boldsymbol{\alpha}$ to the trade-off space, as shown in black. 2) Starting from each model, we scaled the feature and the sensor sets $\mathbf{F}$, $\mathbf{S}$, as shown in blue. 3) We then scaled the precision $nb$ of each of these pruned configurations (shown by the grey curves) and we finally extracted the Pareto front shown in red. As shown by the Pareto configurations highlighted in green, our method preserves the highest baseline train-set accuracy by pruning 11 of the available 28 features, which results in $C_{HA}$ savings of 53%. When willing to tolerate further accuracy losses of 0.4%, our method outputs a configuration that consumes only 13% of the original cost by using a smaller model ($\alpha_3$), pruning 18 features, turning one sensor off and using a 32 bit representation. Figures 3(c,d) break down the computational cost $C_{AC}$ and the sensor costs $C_{SI}$. When considering only the costs of the AC evaluation (graph (c)), our method results in savings of almost two orders of

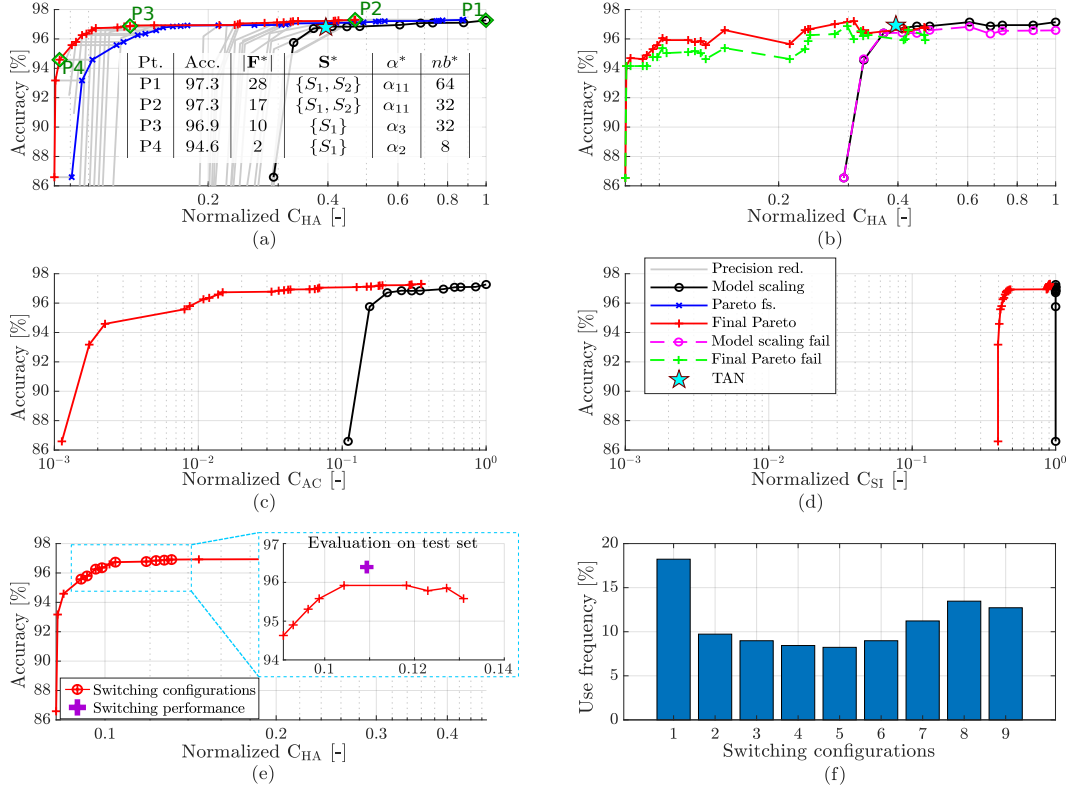

Figure 3: Experiments on the Human Activity Recognition benchmark.

magnitude with accuracy losses lower than $1\%$, and up to 3 orders of magnitude when tolerating more losses. Indeed, these computational cost savings result from the joint effects of precision reduction and of simplifying the AC with Algorithm 2, with savings of about 2 to 10 % per feature pruned (see Appendix B.2). Sensor and feature costs, as shown in graph (d) only scale up to 50%, since at least one of the sensors must always be operating. This demonstrates the importance of taking these costs into account: even though computation costs savings are impressive, the system is still limited by the sensing costs.

**Robustness and online deployment.** The red curve in Figure 3(b) shows that the evaluation of the selected Pareto configurations against testing data stays within a range of $\pm 1\%$ with respect to Figure 3(a). Comparing our method to the TAN classifier denoted with the cyan marker, we can see that it provides further cost saving opportunities, while achieving competitive accuracy. We also assessed the robustness of our method by simulating, per configuration, ten iterations of random failure of varying sizes of feature sets ($|\mathbf{F}|/10,|\mathbf{F}|/5,|\mathbf{F}|/2$). The green and magenta dotted curves show the median of these experiments for the Pareto configurations and for the original model set. These trials stay within a range of $-2\%$ compared to the fully functional results in red and black, which validates our choice of a generative learner that can naturally cope with missing features at prediction time.

In embedded sensing scenarios, environmental circumstances, power consumption requirements and accuracy constraints commonly vary over time. This calls for dynamic operating settings, where the system can switch between different accuracy-cost operating points at run-time. Figure 3(e) shows such a scenario for nine operating points selected off-line (as highlighted with circular markers in the background graph), which comply with hypothetical user needs of accuracy>95% and normalized cost<20%. The implemented policy assumes an energy efficient mode when the classifier has recently identified that there is no ongoing activity, and a high reliability – and costlier – mode when it has identified that there is ongoing activity (see Appendix B.3 for more details). The foreground of Figure 3(e) contrasts the test-set cost-accuracy performance attained when always using the same model (in red), with the cost-accuracy performance resulting from the implementation of our model-switching policy (purple cross). Even with its simplicity, the proposed policy attains accuracy vs. cost improvements that go beyond the static Pareto front. Figure 3(f) shows that this

Table 3: Results for benchmarking datasets [$C_{AC}$,$Acc_{tr}$%,$Acc_{te}$%].

| Dataset | Operating pt. 1 | Operating pt.2 | Operating pt. 3 | Operating pt. 4 | TAN |
|---------|-----------------|----------------|-----------------|-----------------|-----|
| Banknote | [1,94.5,95.6] | [0.09,94.5,95.6] | [0.04,89.9,93.1] | [0.01,84.5,86.8] | [0.24,93.8,92.2] |
| Houses | [1,97.6,97.4] | [0.12,97.6,97.3] | [0.04,97.1,96.6] | [0.01,94.3,94.0] | [0.05,97.2,97.1] |
| Jester | [1,75.6,76.4] | [0.35,75.6,76.4] | [0.12,74.7,75.7] | [0.02,72.6,73.1] | [0.12,73.1,72.3] |
| Madelone | [1,68.1,68.4] | [0.05,68.6,69.1] | [0.02,66.9,68.8] | [0.01,62.6,62.9] | [0.13,66.0,65.7] |
| Nltcs | [1,93.5,93.9] | [0.19,93.6,93.8] | [0.03,93.4,94.2] | [0.01,91.7,92.0] | [0.11,91.4,91.9] |
| Six-HAR | [1,91.5,89.8] | [0.38,91.6,89.9] | [0.15,89.3,89.3] | [0.04,89.8,89.8] | [0.36,91.7,90.3] |
| Wilt | [1,97.1,97.5] | [0.07,97.1,97.5] | [0.03,97.1,97.5] | [0.01,96.9,97.5] | [0.25,97.1,97.5] |

is achieved by making a balanced use of the nine available configurations. Note that this switching action incurs overhead only in terms of memory since the set of Pareto switching configurations is always determined off-line, and will be only fetched when needed. In most portable applications, predictions must be made at a much higher frequency than configuration changes are necessary [12]. The incurred memory overhead in our experiments is less than 3% of the total cost, since model switching is only necessary on 120 out of the 1470 predictions.

## 6.2 Generality of the method: evaluation on benchmark datasets

We now apply our optimization sequence to the datasets in Table 1. For lack of information on the hardware that originated these datasets, we only consider the computation cost $C_{AC}$, again evaluated on the cost model of the ARM M9 CPU. Table 3 shows this cost along with the training and testing accuracy ($Acc_{tr}$,$Acc_{te}$) at four operating points for every dataset. Note that we have also included the six-class HAR benchmark, to demonstrate the applicability of our method beyond binary classification. We can see that all the benchmarks strongly benefit from our proposed methodology, that they are all robust when contrasted against the test dataset, and that they are competitive when compared to a TAN classifier. Appendix A shows a figure with the Pareto fronts for all the experiments herewith.

## 7 Related work

The problem of hardware-efficient probabilistic inference has been addressed by the probabilistic-models and the embedded-hardware communities from several perspectives. The works by Tschi-atschek and Pernkopf [40] and Piatkowski et al. [32] propose reduced precision and integer representation schemes for PGMs as a strategy to address the constraints of embedded devices. In [36], Shah et al. propose a framework that automatically chooses an appropriate energy-efficient low precision representation and generates custom hardware. Other efficient hardware implementation efforts have been made by Zermani et al. [43], Schuman et al. [35], and Sommer et al. [37], who have proposed to accelerate inference on SPNs, capitalizing on their tractable properties.

Our work constitutes an effort to integrate the expertise from both communities under a unified framework, which considers the impact of all scalable aspects of the model to optimize it in a hardware-aware fashion. To that end, it leverages the properties of the selected AC representation. Such representation enables the use of our framework with any probabilistic model that is compute-efficient at prediction time: see [17] by Holtzen et al. and [41] by Vlasselaer et al. for examples of probabilistic program compilation to ACs; and [28] by Lowd and Rooshenas on how to perform efficient inference with Markov networks represented as ACs.

## 8 Conclusions

We proposed a novel hardware-aware cost metric to deal with the limitations of the efficiency vs. performance trade-off considered by the field of tractable learning. Our method obtains the Pareto-optimal system-configuration set in the hardware-aware cost vs. accuracy space. The proposed solution consists of a sequential hardware-aware search and a Pareto-optimal configuration selection stage. Experiments on a variety of benchmarks demonstrated the effectiveness of the approach and sacrifice little to no accuracy for significant cost savings. This opens up opportunities for the efficient implementation of probabilistic models in resource-constrained edge devices, operating in dynamic environments.

## Acknowledgements

We thank Yitao Liang for the LearnPSDD algorithm and for helpful feedback. This work is partially supported by the EU ERC Project Re-SENSE under Grant ERC-2016-STG-71503; NSF grants #IIS-1633857, #CCF-1837129, DARPA XAI grant #N66001-17-2-4032, NEC Research, and gifts from Intel and Facebook Research. This research received funding from the Flemish Government under the "Onderzoeksprogramma Artificiële Intelligentie (AI) Vlaanderen" programme.

## Footnotes

[1]Conditional probability can also be performed by an upward and a downward pass [6].

[2]In this work we assume that the local cache size is sufficiently large to store intermediate values, but not large enough to store parameters. However, for some learned circuits, there are about as many parameters as edges, so depending on the local memory size, one might need to store intermediate values also in main memory.

[3] In Alg. 1 the user can provide the desired accuracy or cost as the while-loop break criterion.

[4]Code available at https://github.com/laurago894/HwAwareProb.

[5]Using the ACE compiler available at `http://reasoning.cs.ucla.edu/ace/`.

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
