[Supplementary Material]

# A. Experiments on benchmarking datasets

Cost vs. accuracy mapping for all datasets: Pareto curves obtained by the proposed method on the left, and their evaluation on the test dataset on the right. The black line represents the models complexity scaling step, the blue one represents the sensor interface scaling stage and the red one the final Pareto optimal configuration set. The cyan star corresponds to the performance of a Tree Augmented Naive Bayes (TAN) classifier after compilation to Arithmetic Circuit representation, and the green markers indicate the data points reported in the paper.

## B. Experiment on the benchmark HAR

### B.1 Sensor and feature costs

We assume that feature extraction and classification take place in an ARM 9 CPU, and we thus define these costs relative to computational costs.

**Sensor costs.** We assume that feature extraction and classification take place in the aforementioned CPU, which consumes, on average, 1 Watt to execute 10G operations per second. It will thus consume approximately 0.1nJ per operation of the CPU.[1] Considering that the largest AC requires about 20000 operations per instance classified, we assume a total computational energy consumption of $2\mu$J. We assume that both the gyroscope and the accelerometer consume at least 2 mW when operating at 10KHz, and that they thus consume roughly $0.2\mu$J per operation of the CPU.[2] Thus, we set the cost of the sensors relative to the computation costs: each sensor has a cost 10% the total cost of classifying a single instance in the most complex model available.

**Feature costs.** The features of this dataset are extracted by sampling the sensory signal, applying three low-pass filters and calculating statistical quantities (mean,maximum/minimum,correlation and standard deviation) on the resulting signal. Sampling and extracting the statistical features require a small number of operations in comparison to filtering. For example, calculating the mean of a sample requires a single MAC (Multiply-Accumulate, consisting of a multiplication and an addition) operation, whereas a 3rd order low pass filter will require at least nine. Filtering thus takes the bulk of the computations, so we assume that each feature extraction incurs in a cost of 30 MAC operations.

## B.2 Computational costs savings per feature pruned

The AC pruning strategy of Algorithm 2 is driven by feature selection, i.e. every time a feature is pruned, the model can be reduced by pre-computing operations related to unobserved indicator leaf nodes. The figure below shows an analysis of global $C_{AC}$ savings per feature pruned, for every model used in the experiments.

The histogram shows local $C_{AC}$ savings: in most cases, every time a feature is pruned, $C_{AC}$ is reduced by 1% to 5 %, and up to 13%.

Distribution of $C_{AC}$ savings per feature pruned

## B.3 Model switching experiment

The Pareto-configuration model set used in this experiment was selected according to hypothetical user's needs: "accuracy must be larger than 95% and the normalized cost lower than 20%".

The policy we proposed, assumes that classifying "no activity" is a lower complexity and lower stakes task and therefore gives priority to the task of classifying "activity": The model remains fixed from $t-11$ to $t-1$. At time $t$, the policy switches to a more complex model when the prediction is "activity", and it switches back to a simpler model when the prediction is "no activity".

## B.4 Features used

1 absolute energy-acc.x-value1
2 absolute energy-acc.x-value2
3 absolute sum of changes-acc.x-value1
4 absolute sum of changes-acc.x-value2
5 maximum-acc.x-value1
6 maximum-acc.x-value2
7 maximum-acc.x-value3
8 skewness-acc.x-value4
9 standard deviation-acc.y-value1
10 standard deviation-acc.y-value2
11 standard deviation-acc.y-value3
12 sum values-acc.y-value1
13 count above mean-acc.z-value1
14 count below mean-acc.z-value1

15 mean change-acc.x-value1
16 longest strike above mean-acc.z-value1
17 longest strike below mean-acc.z-value1
18 maximum-gyro.z-value1
19 median-gyro.z-value1
20 median-gyro.z-value2
21 median-gyro.z-value3
22 median-gyro.z-value4
23 number crossing-acc.z-value1
24 standard deviation-gyro.z-value1
25 median-acc.x-value1
26 median-acc.x-value2
27 standard deviation-gyro.x-value1
28 standard deviation-gyro.y-value1

## Footnotes

[1] https://developer.arm.com/products/processors/classic-processors

[2] Gyroscopes can consume as much as 10 times more energy than accelerometers, but we can assume that they are both part of a larger system, such as a Inertial Measurement Unit, hence we assign the same cost to both.