[Reviews · NeurIPS 2019]

Reviewer 1



The authors propose a method to trade-off "computational costs" and "model fit" when learning a Sum-Product-Network (SPNs) represented as an Arithmetic Circuit. An SPN is a compact representation of a probabilistic model over discrete random variables with finite domain. The proposed method involves an SPN learner that is restricted to binary random variables. In practice, this requires to convert continuous variables into categoricals (e.g., using binning), and categoricals into binaries. While SPNs can handle missing data, they do are typically black-box models where the structure is learned. Computational costs are defined in terms of costs per arithmetic operations, memory/caching costs, as well as costs for feature computation. In particular, the authors assume that the feature costs might be defined over groups of features, e.g., generated by a single sensor. The made choices to define costs are plausible and rather straightforward. Model fit is measured in terms of identifying a distribution's mode, i.e., predictive performance of a maximum likelihood estimate. While the authors include datasets for density estimation in the experiments, they still focus on predicting one of the attributes rather than estimating the density of the whole data. While the proposed approach can be applied to a large and important class of problems, the title and introduction slightly overstate its applicability. The method does not address a general probabilistic model, e.g., stated as a Markov network or a probabilistic program. The significance of the work is moderate. The main contribution, besides defining hardware-related costs, is a heuristic to identify a set of SPNs at the Pareto frontier of costs vs. performance. The heuristic performs a backwards feature selection, where features are greedily removed such that the ratio of accuracy and costs is maximized. The removal is performed by pruning a previously trained SPN using an existing SPN learning algorithm. While empirical experiments confirm the effectiveness of the proposed search heuristic, the method is similar to backward feature selection sequentially selecting features that least affect the performance-cost ratio. The overall scientific contribution to the field of Machine Learning is minor. The pseudo code in Algorithm 1 is, in parts, incorrect: a_{ca,j}, acc_{ca,j}, cost_{ca,j} should be outside of the S-loop; \alpha_{select} etc. should be outside of the j-loop. Also using \notin for two sets, F_{ca,j} and F_S, is imprecise; this should be F_{ca,j} \cap F_S = \varnothing. Finally, \argmax_{F \notin F_{ca}} is ambiguous as there might be many F that are not element of F_{ca} (which is, strictly speaking, a list of sets). I assume, this should read j^* = \argmax_j CF(acc_{ca,j}, cost_{ca,j}), \alpha_select = \alpha_{ca,j^*}, and F_{rm} = \F_{ca,j^*}. Overall, I find the pseudo code not very helpful in its current form. Figure is slightly too small.

Reviewer 2



-- I enjoyed reading this paper, it is well-written, and tackles an important problem. -- My main concern is around the overheads added by the proposed algorithms. First of all, it wasn't clear from reading the paper if these algorithms need to be run for each inference query. Please clarify. If so, even though the complexity analysis is given for different components, it will help if this was empirically shown too. For instance, for each of the datasets, and inference tasks, what was the breakdown of the time spent in choosing a configuration, and actually serving a prediction for these tasks? -- The key motivation of the proposed work is for the edge computing use cases, which are as noted in the paper, latency and privacy sensitive. Thus, the overhead analysis becomes even more important. -- The paper has mainly used accuracy as the performance metric, and mentioned that it could be any other application-specific performance metric. However, it isn't clear how. Applications may have their own metrics, and they could vary over time as the paper points out too. Hence, it would definitely strengthen the practical aspects of the approach to show this aspect more clearly. -- nits: --- What is column 2 in Table 2? The heading says "64-bits". Is this the actual cost computed for 64-bits for each of the operations? Please consider renaming the header for this column if so. --- Fig. 3 is microscopic :). Please consider devoting more space. === UPDATE After Reading The Author Response === I would like to thank the authors for a great response, it clarified and satisfied my questions about the overheads and performance metrics. I am happy to maintain my current score of 7.

Reviewer 3



# Summary The paper proposes a resource-aware cost metric that takes into consideration the target embedded device's properties and system-level configuration, thus introducing probabilistic reasoning on edge computing. # Quality The proposed approach has been evaluated on different datasets for classification tasks. The paper should stress that the methods works only for classification tasks and the generality of the AC as a density estimator is lost. Indeed, since the methods relies on a greedy feature selection approach, queries on removed variables cannot be answered. # Clarity The authors firstly provide how to compute the computation cost of an AC in terms of arithmetic operations and fetching parameters from off-chip memory and storing and fetching from local cache. They the introduce the sensor interfacing cost. A greedy algorithm for learning the lowest-cost/highest-accuracy AC has been proposed that iteratively tries to remove a feature taking into account the accuracy of the new obtained AC. It is not clear the relationship between the computation cost C_AC and the cost of extracting features C_F. It seems they are strongly correlated. For instance, the gain in C_AC with respect to the each removed feature could be interesting, at least empirically. # Originality The work proposes a new approach on learning lowest-cost AC classifiers with high accuracy. The related section should be extended. See for instance [1] and [2] that are cited in the paper. Differences wrt these works should be indicated. [1] Integer undirected graphical models for resource-constrained systems, 2016 [2] Automatic mapping of the sum-product network inference problem to fpga-based accelerators, 2018 # Significance The experimental evaluation of the proposed approach prove its validity. However, there is no comparison with respect to other approaches. For instance, at least a base comparison to the accuracy obtained with the tree-augmented BN used for feature selection should be reported in the table. Minors line 32: remove the ";" line 113: represents --> represent line 115: defines --> define line 120: benchmarks,customized --> benchmarks, customized

[Author Response · NeurIPS 2019]

We would like to thank the reviewers for their thoughtful comments and valuable suggestions.

**[R1,R3] Using our techniques for density estimation, not classification:** our techniques for complexity scaling during learning, numerical precision reduction and for finding the Pareto optimal set of configurations apply directly to the scenario where the goal is density estimation. But as R3 notes, the feature and sensor pruning operation is not applicable if one wants to keep the full joint distribution: for example, the black line in Figure 3(b) shows only model complexity scaling in terms of computation cost. The paper focuses on classification because it addresses restrictions of embedded sensory applications, like in the smartphone-based human activity recognition benchmark. However, having a joint distribution available is still an essential property of ACs/SPNs, and keeps them robust to missing data from unavailable sensors as shown in Figure 3(d) (green and magenta lines).

**[R1] Applying this more broadly to Markov networks or probabilistic programs:** Our method applies to all probabilistic models that are compute-efficient at prediction time, because such models can be compiled into circuits. Please see the UAI-2019 tutorial slides on Tractable Probabilistic Models by Van den Broeck, Vergari and Di Mauro for examples of how to compile PGMs (see also example below) and probabilistic programs into circuits. In fact, the state of the art for discrete probabilistic program inference is to compile into tractable circuits, where our techniques become directly applicable (see work of Holtzen, Vlasselaer, Riguzzi, etc.). We will clarify this point in the paper.

**[R1] Being "restricted to binary random variables":** All the datasets in our experiments include multi-valued features: we introduce a binary indicator variable for each of their values. Such a one-hot encoding is standard in SPN learning and deep learning, and is w.l.o.g.; thus, our approach supports categorical variables already. In principle, by using another type of SPN learner, our method could also support continuous random variables, which can be modeled using univariate Gaussian leafs in the circuit. Our algorithms are agnostic to the leaf distributions used.

**[R1] About "the method is similar to backward feature selection":** There is a lot more going on in our algorithm: precision scaling, model simplification, hardware-aware prediction, cost measurements, etc. In addition, pruning features directly impacts computation and sensory interfacing costs, as noted by R3 and discussed more below.

**[R1] Improving the pseudocode:** Thanks for this valuable feedback, we will improve the pseudocode as you suggest.

**[R1] Including datasets for density estimation:** We considered datasets from both the probabilistic models and the general machine learning communities, to demonstrate that our method is effective regardless of the data's nature.

**[R2] Runtime overhead:** The set of Pareto configurations is always determined off-line, and each resulting model subsequently saved in memory. The processor will fetch and execute them at run-time according to accuracy and cost needs or to comply with a given policy. As such, there is memory overhead but no computational overhead. In addition, predictions must be made at a much higher frequency than configuration changes are necessary in most applications, so there is no need to run the optimization for each inference query. In the on-line deployment experiment of Section 6.1, the total energy-cost overhead is of less than 5% of the prediction cost. We will include this breakdown in the paper.

**[R2] Using other performance metrics:** Other application-specific performance metrics can replace the accuracy terms in Algorithms 1 and 3 and can be modified as needed (e.g. medical applications often aim to balance precision and recall, and may favor the latter at night under scarce medical supervision). We will clarify this in the paper.

**[R3] On the correlation between computation and feature cost:** Thank you for highlighting this interesting correlation, which is induced by the AC pruning component of our search strategy. We will comment on this aspect and include an empirical analysis: e.g. for the HAR benchmark, we have seen a reduction on $C_{AC}$ of 4 to 8 % per feature pruned.

**[R3] Extending the related section:** Thanks for the valuable feedback, we will modify the related section as suggested.

**[R3,R1] Comparison to other approaches:** On the left side of the table below, we report the performance of a Tree Augmented Naive Bayes classifier after its compilation to an AC ($\text{Acc}_{\text{te}}, C_{AC}$). We will add these results to the paper, as they show how to apply PGMs to our framework; and that PSDDs have a superior performance on most benchmarks.

**[R1] Being restricted to binary classification** We will include the 6-class HAR experiment reported in the table below, which proves our method works with multi-class problem and that it is comparable to the TAN baseline.

| Banknote | HAR | Houses | Jester | Madelone | Nltcs | Wilt | HAR multi-class Pareto / TAN |
|---|---|---|---|---|---|---|---|
| 92.15, 0.5 | 96.9, 0.5 | 97.09, 0.1 | 72.25, 0.2 | 65.72, 0.3 | 91.96, 0.2 | 97.5, 0.6 | 91.2,1; 91,0.42; 88,0.3 / TAN: 91,0.4 |

**[R2] The heading "64-bits in Table 2 is unclear:"** We thank R2 for noting this, we confirm that it refers to the cost computed for that precision, and we will rename the headers to make this clearer.

**[R1,R2,R3] Typos, errors and format changes:** We thank R1 for identifying the errors of Algorithm 1 and R3 for identifying several typos. We will make the due corrections. We also thank R1 and R2 for pointing out that Fig. 3 is too small, we will assign it due space.

[Meta-Review · NeurIPS 2019]

The authors present an interesting contribution to sum-product networks and yet there are some concerns on its impact and significance, hence the mixed reviews.